# Validity of ICD codes to identify do-not-resuscitate orders among older adults with heart failure: A single center study

**Katherine Callahan**[ID]◉*, **Yubraj Acharya**[ID]◉, **Christopher S. Hollenbeak**◉

Department of Health Policy and Administration, College of Health and Human Development, The Pennsylvania State University, University Park, Williamsport, Pennsylvania, United States of America

◉ These authors contributed equally to this work.

* klc83@psu.edu

## Abstract

### Background

Observational research on the advance care planning (ACP) process is limited by a lack of easily accessible ACP variables in many large datasets. The objective of this study was to determine whether International Classification of Disease (ICD) codes for do-not-resuscitate (DNR) orders are valid proxies for the presence of a DNR recorded in the electronic medical record (EMR).

### Methods

We studied 5,016 patients over the age of 65 who were admitted to a large, mid-Atlantic medical center with a primary diagnosis of heart failure. DNR orders were identified in billing records from ICD-9 and ICD-10 codes. DNR orders were also identified in the EMR by a manual search of physician notes. Sensitivity, specificity, positive predictive value and negative predictive value were calculated as well as measures of agreement and disagreement. In addition, estimates of associations with mortality and costs were calculated using the DNR documented in EMR and the DNR proxy identified in ICD codes.

### Results

Relative to the gold standard of the EMR, DNR orders identified in ICD codes had an estimated sensitivity of 84.6%, specificity of 96.6%, positive predictive value of 90.5%, and negative predictive value of 94.3%. The estimated kappa statistic was 0.83, although McNemar's test suggested there was some systematic disagreement between the DNR from ICD codes and the EMR.

**Data Availability Statement:** There are legal restrictions to share the de-identified data as it came from billing records and electronic medical records from Penn State Milton S. Hershey Medical

Center. These restrictions have not been imposed by the Penn State Institutional Review Board, but by Penn State Milton S. Hershey Medical Center. Contact information regarding data access is: Cletis Earle - Chief Information Officer - Penn State Health (E-mail: cearle1@pennstatehealth.psu.edu).

**Funding:** The author(s) received no specific funding for this work.

**Competing interests:** The authors have declared that no competing interests exist.

## Conclusions

ICD codes appear to provide a reasonable proxy for DNR orders among hospitalized older adults with heart failure. Further research is necessary to determine if billing codes can identify DNR orders in other populations.

## Introduction

Research on issues in advance care planning (ACP) can be challenging as there are few ACP variables routinely collected in large datasets [1, 2]. Some datasets, such as the Health Retirement Study's (HRS) Exit Interview, which began conducting surveys in 1996, contain survey data on advance care planning. However, the data collected in the Exit Interview are completed with a "proxy informant" of HRS participants who have died, and the number of cases with this information is relatively small with less than 1,000 participants per year [3]. The ability to identify ACP data in large administrative and other observational health care datasets that are easily accessible to health services researchers would substantially help advance research in ACP.

One potentially important source of data is administrative discharge data, which are collected by hospitals that use them for billing, monitoring, and reporting to organizations that measure quality of care [4]. These data provide information on hospital stays and routinely include International Classification of Diseases (ICD) codes, which reflect patient diagnoses and procedures patients receive during their hospitalization [4]. Another potential source of data is claims data, which records information from third-party payers and often includes ICD codes to document diagnoses and procedures. Since 2010 there has been an ICD code for do-not-resuscitate (DNR) orders—V49.86 in the 9th revision and Z66 in the 10th revision, which began to be widely used in the US in October 2015 [1, 5]. A DNR is a signed medical order that tells healthcare providers not to begin cardiopulmonary resuscitation (CPR) in the event of cardiac arrest [6]. In recent years, the documentation of a DNR status has been incorporated into all-encompassing legal documents known as Medical Orders for Life Sustaining Treatment (MOLST). These forms include options for other forms of life sustaining treatments such as breathing machines and feeding tubes, and regardless of how DNR status is documented for patients they are still recorded in patient charts or electronic medical records (EMR) [7]. Although a DNR is only part of the ACP process, it can be important for patients to define their resuscitation preferences as CPR is often not beneficial for individuals at end-of-life (EOL) who are already weak and frail, such as those with severe chronic and terminal illnesses such as heart failure (HF) [8]. Almost half of individuals die within the first five years of a HF diagnosis and it has an exacerbation-prone trajectory that often results in frequent hospitalizations, which contrasts with chronic, declining trajectories more common in other chronic and terminal illnesses such as dementia [9]. Researchers therefore have access to a proxy for DNR orders through the widely available ICD codes. This proxy may allow for research around DNR orders as part of the ACP process. For example, we recently used ICD-9 codes as a proxy for DNR orders to examine whether patients with DNR orders were more likely to be readmitted after being hospitalized in Pennsylvania for HF [10].

A major concern about ICD codes for DNR orders is it is not known whether they are valid proxies for actual DNR orders, and whether they reflect a patient's resuscitation preference that is recorded in their medical record. There are several reasons that the ICD code may not reflect a patient's actual DNR status. ICD codes are used by hospital billing coders in order to maximize revenues to the hospital. If the ICD code for DNR does not have revenue value in billing then hospital billing coders may not prioritize recording DNR in the limited slots

available for entry. Under this scenario, DNRs could be routinely under-represented in administrative data. If the ICD code for a DNR order is not a reasonably accurate measure for the legal document that a patient signs to make their wishes regarding CPR known to the clinical team, then caution should be used in using it as a proxy for DNR, and previous studies that have used it should be interpreted with caution. However, if the ICD codes could be shown to be valid proxies for actual DNR status, then observational data sets such as administrative discharge data and claims data may be useful for EOL research.

There is some evidence that the ICD code for the DNR order is accurate [1, 6]. For example, one study found that that the ICD-9 code for the DNR order (V49.86) is moderately sensitive and highly specific among a cross-section of patients admitted to a hospital [1]. Another study that investigated the validity of the ICD-9 code among primary diagnoses of myocardial infraction, pneumonia, and heart failure (HF), suggested that the DNR proxy has a high sensitivity but only moderate specificity [6]. While these studies suggest that the ICD code provides a potentially promising proxy for DNR, there are several limitations. These studies include primary diagnoses that include many observations for which a DNR order is not likely to be relevant. For example, it is reasonable to think that discussing resuscitation preferences with a 16-year-old patient admitted with a distal radius fracture is not a high priority for health care providers. This in turn can affect the accuracy of the ICD code documented. In addition, these studies did not include data on the more recent ICD-10 codes that have been in widespread use in the US since 2015.

In order to address these gaps, this study sought to determine the validity of ICD proxies for DNR in elderly patients hospitalized with a primary diagnosis of HF. Our sample includes both ICD-9 and ICD-10 codes for DNR, and also explores whether use of the ICD proxy produces different estimates when used as a substitute for a gold standard measure of DNR orders found in the EMR.

## Objectives

The purpose of this study was to determine whether ICD-9 and ICD-10 codes for DNR are valid proxies for the presence of a DNR that is recorded in the EMR. We utilized data from a large, academic hospital in the mid-Atlantic region and merged billing records that contain ICD codes with data regarding DNRs documented in the EMR. In addition to comparing ICD DNR proxies to actual DNR documentation, we also estimated the association between DNR and mortality and costs using both the ICD code DNR and the DNR recorded in the patients EMR in order to determine whether the ICD measure provides similar estimates of association relative to actual DNR documented in the EMR.

## Materials and methods

### Data source

This research used data from the Penn State Milton S. Hershey Medical Center. Billing records were obtained from the cost-accounting system (Change Healthcare, Nashville, TN), which provided ICD codes from which DNR was identified, as well as details on patient characteristics, hospital stay, and total costs. In addition, the EMR for each patient was examined, which provided physician notes that documented the patient's treatment preferences, including a DNR order. This study was reviewed and approved by the Pennsylvania State University Institutional Review Board and informed consent was not required. Data were received de-identified from the medical center's decision support department. In addition, two authors (KC and CH) had full access to all the data in the study and take responsibility for its integrity and the data analysis.

## Cohort

Patients admitted between 2013 and 2018 with a principle diagnosis of HF were identified using the ICD-9 code of 428.xx (heart failure) and ICD-10 code of I50.xx (heart failure). Patients under the age of 65 were excluded as older patients are more likely to participate in any type of ACP [11, 12]. The final sample included n = 5,016 patients.

## Outcomes

DNR proxies in ICD codes (iDNR) were identified using an ICD-9 code of V49.89 and an ICD-10 code of Z66 among secondary diagnosis codes in billing records. The actual DNR contained in the EMR (eDNR) was identified by searching the EMR, particularly physician notes, for any mention of DNR status. Secondary outcomes included in-hospital mortality and total cost of admission. Costs are estimated in the cost-accounting system using ratios of costs-to-charges at the department level and include an overhead allocation. Thus, total costs in this study represent fully-loaded operating costs for the HF admission. Furthermore, costs were adjusted for inflation to 2021 US dollars using the medical care component of the Consumer Price Index.

## Covariates

Several covariates that have been shown to be associated with DNR in the existing literature were included in the analyses. To control for patient characteristics, indicators were used for age groups (65–74, 75–84, or 85+), sex (male or female), race (white or other race/ethnicity), payer (Medicare, Medicaid, commercial, or other), and hospital admission type (elective, emergent, or urgent). To control for comorbidities, we included the Charlson comorbidity index (CCI), which is a weighted sum of seventeen common comorbidities that can be identified using ICD codes [13–15]. The CCI was transformed into four categories (0, 1, 2–3, or 4+).

## Statistical analysis

In order to determine whether the iDNR predicts the presence of eDNR, the sensitivity and specificity of the iDNR were estimated relative to the eDNR as the gold standard. Sensitivity is defined as the probability that a patient has an iDNR given that patient has an eDNR; specificity is defined as the probability that a patient does not have an iDNR given that the patient does not have an eDNR. We also computed the positive predictive value (PPV), which is defined as the probability that a patient has an eDNR given that they have an iDNR, and the negative predictive value (NPV), which is the probability that a patient does not have an eDNR given that they do not have a iDNR. Additionally, these measures were computed across strata of patient characteristics, including age, sex, race/ethnicity, and comorbidities.

Two univariate tests of agreement between the two DNR measures were performed [16]. First, the level of agreement between the iDNR and eDNR was estimated using a Cohen's kappa statistic. The kappa statistic is a correlation measure with values between -1 and 1 that reflect the overall level of agreement between two predictions and is often used to measure inter-rater reliability [17, 18]. Second, systematic disagreement between the iDNR and eDNR was compared using McNemar's test. McNemar's test was used to investigate whether there are systematic differences in disagreement between the two measures. In this case, it tests whether the iDNR is significantly more likely to be negative when the eDNR is positive, or whether the eDNR is more likely to be negative when the iDNR is positive [19].

After estimating agreement between the two DNR measures, we examined whether the use of each measure would yield substantially different estimates of association when used as a

**Table 1. Summary statistics for elderly HF patients, stratified by DNR status in ICD codes and EMR.**

| Variable | No DNR | ICD DNR | EMR DNR |
|---|---|---|---|
| | (N = 3398) | (N = 1388) | (N = 1498) |
| Age | | | |
| 65–74 | 43.73% | 18.23% | 18.89% |
| 75–84 | 37.32% | 34.08% | 33.71% |
| 85+ | 18.95% | 47.69% | 47.40% |
| Sex | | | |
| Female | 43.11% | 55.12% | 54.81% |
| Male | 56.89% | 44.88% | 45.19% |
| Race | | | |
| White | 93.76% | 96.76% | 96.93% |
| Other Race | 6.24% | 3.24% | 3.07% |
| Charlson Comorbidity Index Score | | | |
| 0 | 63.83% | 63.33% | 60.21% |
| 1 | 7.65% | 7.49% | 8.48% |
| 2 to 3 | 17.30% | 16.79% | 17.49% |
| 4+ | 11.21% | 12.39% | 13.82% |
| Payor | | | |
| Commercial | 19.63% | 16.86% | 16.69% |
| Medicare | 70.95% | 71.97% | 72.50% |
| Other | 9.42% | 11.17% | 10.81% |

covariate in multivariable analysis about the associations between DNR and outcomes. The outcomes that we were specifically interested in—mortality and costs—were modeled using the iDNR and eDNR as independent variables and controlling for other covariates. Finding that the two measures yielded substantially different associations would be evidence that the iDNR is not a good proxy for eDNR. Linear probability models were used to model the relationship between DNR and mortality, after controlling for the covariates described above. Linear regression models were used to model the association between total costs and DNR after controlling for the patient characteristics. Stata software (version 15.1, College Station, TX) was used for all analyses, and statistical significance was defined as $p < 0.05$.

## Results

### Patient characteristics

A description of patient characteristics stratified by DNR status is provided in Table 1. The summary compares elderly HF patients without a DNR (n = 3,398) to those with an iDNR (n = 1,388) and those with an eDNR (n = 1,498) across several covariates. In this sample, elderly HF patients were more likely to have a DNR (either iDNR or eDNR) if they were older, white, female, and had Medicare as their primary insurance. There were no substantial differences between patients with an iDNR and patients with an eDNR. The largest difference observed was between CCI categories, where there were slightly more patients with an iDNR that had a score of zero compared to patients with an eDNR (63% vs. 60%). Compared to patients with DNRs, patients without a DNR were more likely to have a CCI score of one to three, yet slightly less likely to have a CCI score of zero or four or more.

**Table 2. Contingency table of eDNR and iDNR.**

| | | eDNR | | |
|---|---|---|---|---|
| | | *DNR* | *No DNR* | *Total* |
| **iDNR** | *DNR* | 1,268 | 120 | 1,388 |
| | *No DNR* | 230 | 3,398 | 3,628 |
| | *Total* | 1,498 | 3,518 | 5,016 |

## Measures of agreement

A 2×2 contingency table of iDNR by eDNR is shown in Table 2. Cohen's kappa statistic for agreement between iDNR and eDNR was estimated to be 0.83, which suggests "almost perfect agreement" between the iDNR and eDNR [17]. On the other hand, the McNemar test suggested there were systematic differences in disagreement between iDNR and eDNR. In terms of disagreement, a false negative iDNR was significantly more likely than a false positive iDNR. However, as seen in Table 2, the proportion of both false negatives and false positives was small.

## Sensitivity and specificity

For the full sample (N = 5,016), sensitivity of iDNR was estimated to be 84.6%, which means that 84.6% of the individuals who have an eDNR were correctly documented as having an iDNR. Specificity of iDNR was estimated to be 96.6%, which means that 96.6% of patients who did not have an eDNR also did not have an iDNR. The PPV was estimated to be 91.4% and NPV was estimated to be 93.7%, meaning that among the entire sample of individuals with and without DNRs, the iDNR correctly identified who did and did not have a DNR in slightly over 91% of cases. Table 3 shows the sensitivity, specificity, PPV and NPV of the full sample, as well as the sample stratified by patient characteristics of iDNR relative to eDNR. Within the subgroups of patient characteristics, the highest level of sensitivity was found among female patients (87.1%) and the highest level of specificity was found among patients who had a CCI score of 2 to 3 (98.8%). On the other hand, the lowest sensitivity was found among patients who had a CCI score of four or more (79.2%), and the lowest level of specificity was among patients aged 85+ (92.5%).

## Mortality

The iDNR provided similar estimates of the association between DNR and mortality as the eDNR. This is shown in Table 4, which provides the results of linear probability models using both iDNR and eDNR and controlling for covariates. The estimated likelihood of dying before discharge was 11.3 percentage points higher for patients with a DNR as indicated by the iDNR (p<0.001), and 9.3 percentage points higher for patients with an eDNR (p<0.001). Additionally, there were other covariates not related to the central question of the iDNR vs. eDNR that were also associated with in-hospital mortality.

## Hospitalization costs

As seen in Table 5, iDNR and eDNR provided different estimates of the association between DNR and total hospitalization costs, however, they were in the same direction. Linear regression models suggested that, after controlling for covariates and among patients who died, having an iDNR order was associated with cost savings of $19,313 (p = 0.002) and eDNR was associated with cost savings of $39,728 (p<0.001).

**Table 3. Summary of diagnostic characteristics of iDNR relative to eDNR, stratified by patient characteristics.**

| Patient Characteristics | Sample Size | Sensitivity | Specificity | PPV | NPV |
|---|---|---|---|---|---|
| Total | (N = 5016) | 84.60% | 96.60% | 91.40% | 93.70% |
| Age | | | | | |
| 65–74 | (N = 1795) | 80.20% | 98.30% | 95.20% | 92.10% |
| 75–84 | (N = 1815) | 85.30% | 96.80% | 91.90% | 93.90% |
| 85+ | (N = 1406) | 85.90% | 92.50% | 83.00% | 93.90% |
| Sex | | | | | |
| Female | (N = 2336) | 87.10% | 96.70% | 91.80% | 94.60% |
| Male | (N = 2680) | 81.70% | 96.50% | 90.90% | 92.50% |
| Race | | | | | |
| White | (N = 4753) | 84.60% | 96.50% | 91.20% | 93.60% |
| Other Race | (N = 263) | 87.00% | 97.70% | 94.10% | 94.60% |
| Charlson Comorbidity Index Score | | | | | |
| 0 | (N = 3172) | 86.30% | 95.60% | 89.20% | 94.20% |
| 1 | (N = 391) | 78.70% | 98.50% | 95.70% | 91.60% |
| 2 to 3 | (N = 857) | 86.30% | 98.80% | 96.90% | 94.40% |
| 4+ | (N = 596) | 79.20% | 97.90% | 94.30% | 91.70% |
| Code Set | | | | | |
| ICD-9 | (N = 1841) | 81.85% | 98.48% | 95.81% | 92.72% |
| ICD-10 | (N = 3175) | 86.27% | 95.55% | 89.20% | 94.23% |

**Table 4. Results of linear probability models of mortality controlling for eDNR versus iDNR as independent variables as well as other covariates.**

| Variable | eDNR Coefficient | P-Value | iDNR Coefficient | P-Value |
|---|---|---|---|---|
| DNR | 0.093 | <0.001 | 0.113 | <0.001 |
| Age | | | | |
| 65–74 | Reference | | | |
| 75–84 | -0.011 | 0.139 | -0.013 | 0.074 |
| 85+ | -0.022 | 0.007 | -0.027 | 0.001 |
| Sex | | | | |
| Female | -0.014 | 0.024 | -0.015 | 0.014 |
| Male | Reference | | | |
| Race | | | | |
| White | Reference | | | |
| Other Race | -0.012 | 0.375 | -0.012 | 0.383 |
| Charlson Comorbidity Index Score | | | | |
| 0 | Reference | | | |
| 1 | 0.003 | 0.797 | 0.008 | 0.468 |
| 2 to 3 | 0.001 | 0.91 | 0.004 | 0.674 |
| 4+ | 0.018 | 0.071 | 0.022 | 0.023 |
| Payor | | | | |
| Private | -0.002 | 0.772 | -0.003 | 0.726 |
| Medicare | Reference | | | |
| Other | 0.107 | <0.001 | 0.106 | <0.001 |
| Constant | 0.030 | <0.001 | 0.028 | <0.001 |

**Table 5. Results of linear regression models of hospital cost controlling for eDNR versus iDNR as independent variables as well as other covariates.**

| Variable | Coefficient for EMR DNR | P-Value | Coefficient for ICD DNR | P-Value |
|---|---|---|---|---|
| Died | 51,510.77 | <0.001 | 39,331.73 | <0.001 |
| DNR | -668.07 | 0.672 | -1,740.50 | 0.285 |
| Died & DNR | -39,727.87 | <0.001 | -19,312.69 | 0.002 |
| Age | | | | |
| 65–74 | Reference | | | |
| 75–84 | -9,477.56 | <0.001 | -9,346.32 | <0.001 |
| 85+ | -16,159.29 | <0.001 | -16,036.67 | <0.001 |
| Sex | | | | |
| Female | -4,002.47 | 0.003 | -4,008.08 | 0.003 |
| Male | Reference | | | |
| Race/Ethnicity | | | | |
| Non-Hispanic White | Reference | | | |
| Other Race/Ethnicity | -6,017.59 | 0.041 | -6,071.84 | 0.040 |
| Charlson Comorbidity Index Score | | | | |
| 0 | Reference | | | |
| 1 | -12869.1 | <0.001 | -13232.09 | <0.001 |
| 2 to 3 | -8963.363 | <0.001 | -9185.259 | <0.001 |
| 4+ | -9038.276 | <0.001 | -9379.919 | <0.001 |
| Payor | | | | |
| Private | -1,068.03 | 0.531 | -1,170.47 | 0.493 |
| Medicare | Reference | | | |
| Other | -3,096.45 | 0.170 | -3,505.91 | 0.121 |
| Intercept | 36,814.53 | <0.001 | 37,157.06 | <0.001 |

Among patients who died, the mean cost saving for patients with an eDNR and no iDNR was significantly greater than the cost saving for patients with an iDNR and no eDNR. Among patients who died without a DNR, costs were approximately $51,500 higher (p<0.001) in the eDNR model and approximately $39,000 higher (p<0.001) in the iDNR model. Costs were similar in magnitude for patients who had a DNR and survived.

There were other patient characteristics associated with costs. For example, patients who were older, female, non-white, had higher co-morbidity score, and had lower total hospitalization costs. Additionally, compared to those who were insured by Medicare, patients covered by other types of insurance had lower costs of approximately $3,000.

## Discussion

Results from this study suggest that the ICD code for DNR is a valid proxy for the presence of a DNR order documented in the medical record in this sample of elderly HF patients. This was indicated by a high sensitivity (85%), specificity (97%), PPV (91%) and NPV (94%) relative to the gold standard of a DNR documented in the EMR. The estimated kappa statistic ($\kappa = 0.83$) also suggests there is substantial agreement between DNR in billing records and the EMR. It should be noted that the results from McNemar's test suggest a systematic difference between false negatives and false positives; however, this test only considers the off-diagonal elements of the contingency table, which were relatively small compared to the total sample (Table 2). The sensitivity, specificity, PPV, NPV, and kappa statistic all consider the entire contingency table, and therefore together are strong indicators that the iDNR is valid. In comparison to

other common ICD code proxies used in current research studies the iDNR measure performed as well or better [20–22].

However, the estimates of associations between DNR and mortality and DNR and cost using the iDNR and eDNR indicate that when using the iDNR the results need to be carefully considered. While the association between iDNR and mortality and eDNR and mortality were similar (9.3 percentage points vs. 11.3 percentage points), the associations with costs were very different ($19,313 for iDNR, and $39,728 for eDNR). Given the high sensitivity and specificity of iDNR, this finding is most likely due to the fact that there are more false negative results for iDNR, and also because cost data are usually highly skewed and have a large variance. These results suggest caution in using the iDNR as a proxy to measure DNR orders.

This study makes several contributions to the literature on DNR orders and ACP. Although the sample is restrictive, estimates of sensitivity, specificity, PPV, NVP, and agreement suggest that both the ICD-9 and ICD-10 codes are reasonable proxies for DNR orders recorded in elderly HF patient medical records. Second, the results provide confirmation of other validation studies that have examined ICD-9 codes and similarly found that the ICD code is a reasonable proxy for DNR orders [1, 6]. The results of this study indicate that for elderly HF patients estimates of association between iDNR and patient outcomes, such as mortality or costs, are similar in direction to those that would be obtained using eDNR. Additionally, the results of this study support that DNR status can be identified in large administrative databases, enabling researchers to adjust for this as a potential confounder if necessary [23].

Similar to Fonseca et al. (2018), we found the iDNR to have very high specificity at 97%. However, unlike their study, we also found the iDNR sensitivity to be high (85% vs. 69%) [1]. This may be due to differences in the sample, as we only looked at patients over the age of 65 with HF, and the Fonseca et al. study (2018) included patients at any age and admitted with any diagnosis, although the most common admission diagnoses were metastatic cancer, pneumonia, HF, acute myocardial infraction, chronic obstructive pulmonary disease, and stroke. In addition, our iDNR measure included both ICD-9 and ICD-10 diagnosis codes; Fonseca et al. (2018) only included ICD-9 codes in their proxy.

Results of this study differ somewhat from those reported in Goldman et al. (2013) who investigated the accuracy of DNR orders using administrative discharge data in California. They found that DNR orders were more likely to be recorded inaccurately as the time from admission increased. Our study used ICD codes that were determined after the patient was discharged, and includes DNR orders that could have been completed at any time during the hospital stay. One potential explanation for this difference is a difference in the definition of iDNR as DNR orders in California are measured only within 24 hours of admission. They do not capture DNR orders that may have been finalized after admission and may be less accurate for patients with longer hospitalizations [6].

One piece of evidence from our study that suggested that the iDNR might not be a good proxy for eDNR was McNemar's test, which found significant disagreement between the two measures. This test compared the number of false positive results (N = 120) and false negative (N = 230) results and showed that the iDNR yielded systematically more false negatives. Certainly, there should be a greater expectation for false negative results since coders have a limited number of placeholders for ICD codes, and priority is given to those that result in higher reimbursement. The false positive cases are more of a puzzle. In our sample there were 120 false positive cases out of 5,016 patients, which is a relatively small proportion. Their presence raises the question, however, how a billing coder who assigns ICD codes may have indicated the presence of a DNR when we did not find such evidence (i.e., DNR in the EMR). It is possible that these may have just been simple error, it is also possible that DNR orders were counted in false positive cases when patients received a consultation with palliative care or hospice

teams, which they might have interpreted as having a DNR order. Another potential explanation is that coders may have observed evidence of a DNR order for a prior hospitalization and assumed (correctly or incorrectly) that the DNR order would apply to the current hospital admission [6, 24].

## Limitations

The results of this study only came from patients at a single medical center. Therefore, it is possible that the use of ICD codes for DNR orders we observed can be attributed to the practice patterns of this particular center's billing coders as well as the specific practice patterns around ACP and DNR orders of physicians, therefore these results do not reflect practice patterns at other institutions or nationally. It is possible that different coding practices at another hospital would produce significantly different results. Additionally, the patient population at this medical center lacks racial and ethnic diversity, creating the potential for a population with greater racial diversity to have very different results.

This study only examined elderly patients with HF. iDNR may be different for younger patients with HF or for patients with other chronic and terminal illnesses. In the future it may be useful to examine cohorts that include adults aged 40 and older as the use of DNR orders increases at this age [10]. Another limitation to our study the use of only patients with a principal diagnosis of HF. It is possible that the most severely ill patients with HF have a principal diagnosis of acute respiratory failure or myocardial infraction and have HF as the secondary diagnosis [25]. Therefore, it is possible that our sample criteria excluded a part of the HF population.

There are other challenges to measuring associations between patient outcomes and DNR orders that are due to discrepancies between a patient's wishes and care that is actually received. A patient may have a DNR order in place and still receive CPR after a cardiac event. This may occur if the clinical team is unaware of the DNR order or if the patient's family overrides the DNR order. Family members' desire for heroic measures may be given preference to patient's wishes, particularly in hospital environments where lawsuits are more prevalent and if the medical care ends in death for the patient [24]. The greater the extent of these occurrences, the more the estimates of association between DNR orders and outcomes will be biased toward the null.

## Conclusions

Using a sample of elderly patients with a primary diagnosis of HF at a single, large, academic medical center in the mid-Atlantic, this study examined the validity of ICD codes for DNR orders and compared their use relative to DNR status in the EMR. Results showed high sensitivity, specificity, PPV, NPV, and overall agreement. However, they showed that the use of the ICD proxy did not always provide similar estimates of association between patient outcomes and DNR status. These results suggest that the ICD codes provide a reasonable proxy for DNR orders in this sample, and open the door to further applied research on DNR orders. Additional research is needed to validate the ICD proxies using alternative coding pathways to identify HF as discussed above, as well as in other diseases and other populations. In addition, future research should continue to explore other proxies for other elements of ACP beyond DNR orders.

## Author Contributions

**Conceptualization:** Katherine Callahan, Christopher S. Hollenbeak.

**Data curation:** Katherine Callahan, Christopher S. Hollenbeak.

**Formal analysis:** Katherine Callahan, Yubraj Acharya, Christopher S. Hollenbeak.

**Investigation:** Katherine Callahan, Christopher S. Hollenbeak.

**Methodology:** Katherine Callahan, Yubraj Acharya, Christopher S. Hollenbeak.

**Project administration:** Christopher S. Hollenbeak.

**Resources:** Christopher S. Hollenbeak.

**Software:** Christopher S. Hollenbeak.

**Supervision:** Christopher S. Hollenbeak.

**Validation:** Katherine Callahan, Christopher S. Hollenbeak.

**Visualization:** Katherine Callahan.

**Writing – original draft:** Katherine Callahan.

**Writing – review & editing:** Katherine Callahan, Christopher S. Hollenbeak.

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
