## [Decision Letter · Decision Letter 0]

18 Oct 2022

PONE-D-22-21926Validity of ICD Codes to Identify Do-Not-Resuscitate OrdersPLOS ONE

Dear Dr. Callahan,

Thank you for submitting your manuscript to PLOS ONE. After careful consideration, we feel that it has merit but does not fully meet PLOS ONE’s publication criteria as it currently stands. Therefore, we invite you to submit a revised version of the manuscript that addresses the points raised during the review process. Please submit your revised manuscript by Dec 02 2022 11:59PM. If you will need more time than this to complete your revisions, please reply to this message or contact the journal office at plosone@plos.org. Please include the following items when submitting your revised manuscript:A rebuttal letter that responds to each point raised by the academic editor and reviewer(s). You should upload this letter as a separate file labeled 'Response to Reviewers'.A marked-up copy of your manuscript that highlights changes made to the original version. You should upload this as a separate file labeled 'Revised Manuscript with Track Changes'.An unmarked version of your revised paper without tracked changes. You should upload this as a separate file labeled 'Manuscript'.

We look forward to receiving your revised manuscript.

Kind regards,

Keith M. Harris, PhD

Academic Editor

PLOS ONE

Journal Requirements:

3. We note you have included a table to which you do not refer in the text of your manuscript. Please ensure that you refer to Table 2 in your text; if accepted, production will need this reference to link the reader to the Table.

4. Please include your tables as part of your main manuscript and remove the individual files. Please note that supplementary tables (should remain/ be uploaded) as separate "supporting information" files

Additional Editor Comments:

Reviewer 1 has pointed out several important factors to address. Authors are urged to take a more cautious approach to interpreting findings and stating conclusions. Please address each of the reviewer's comments and suggestions. If data is available, suggestions for expanding age range and ethnic diversity, etc. appear reasonable. The treatment of continuous variables and unweighted variables needs to be addressed according to PLOS' requirements for analyses to be conducted according to high standards. This is a valuable work, and we look forward to your response.

Reviewers' comments:

Reviewer's Responses to Questions

**Comments to the Author**

1. Is the manuscript technically sound, and do the data support the conclusions?

Reviewer #1: Partly

2. Has the statistical analysis been performed appropriately and rigorously? 

Reviewer #1: Yes

3. Have the authors made all data underlying the findings in their manuscript fully available?

Reviewer #1: No

4. Is the manuscript presented in an intelligible fashion and written in standard English?

Reviewer #1: Yes

5. Review Comments to the Author

Reviewer #1: Major Points

1. My primary concern regarding the manuscript is the disconnect between the generalizability with which the authors discuss their results and the restrictive nature of their cohort. The authors use their results to discuss how discharge and claims data can be used to identify DNR orders in future studies. However, their cohort is exceedingly restrictive and may not be the most appropriate with which to identify patients admitted with heart failure. Their reported statistics of sensitivity, specificity, PPV, and NPV are only valid for this specific cohort and likely are not generalizable to other conditions or patient cohorts. The authors should first consider their patient selection along the below delineated lines and also discuss their results with much greater clarity about the limitations of their findings throughout the abstract and manuscript, not just in the limitations section of the discussion.

a. Age – the authors select patients who are >65 years of age. Use of DNR orders does increase with age meaning there would be greater prevalence of DNR orders in an older cohort. However, previous studies have indicated use of DNR orders begins to increase at around age 40. Use of an older cohort would result in higher sensitivity and specificity due to the higher prevalence. The authors should address this by either widening the cohort or doing a sensitivity analysis which includes patients down to age 40.

b. Cohort definition – the authors only include patients with a principle diagnosis of HF (ICD9 and ICD10) as the ‘HF’ cohort. However, previous work (Mehta et al. Journal of Critical Care. 2015.) demonstrates that the most severely ill patients with HF (i.e. those most likely to have a DNR order) may have a principle diagnosis of acute respiratory failure and a secondary diagnosis of HF (i.e. presented with hypoxia due to pulmonary edema) or a principle diagnosis of acute MI with a secondary diagnosis of HF. These alterative coding pathways may be used to increase reimbursement but still tend to identify patients with HF. The authors could consider a sensitivity analysis with a wider cohort definition or they should mention that their definition may have selectively excluded individuals most likely to have a DNR order.

c. Choice of heart failure – the authors restrict their analysis to HF only for unclear reasons. They suggest that studies that include all patients may include diagnoses for which DNR may not be relevant. However, DNR orders issued early in the course of admission tend to be more indicative of comorbid condition rather than the acute presenting illness. Moreover, other cohorts such as cancer may be even more relevant for DNR orders. The authors should provide greater justification for why HF was chosen, what biases such a choice may create, and limit their discussion about the utility of their findings to patients with HF.

2. My second major concern deals with the use of a single center and conclusion that can be drawn from it. I commend the authors on their sample size and efforts but I am concerned about any conclusions drawn from a single center on generalizable billing practices that might be heavily biased. The authors correctly mention several possible reasons why DNR orders may be systematically over or under estimated in billing codes from a given center. Each health center tends to bill in a unique fashion. Biases in terms of including or not including certain codes tend to happen at the hospital level, not the patient level. Therefore, the use of a single health system does not necessarily reduce the potential of coding bias to either over or under represent DNR orders in ICD codes. The authors should be far more specific in addressing the limitations of a single center in their discussion. Specifically, they should address the potential for different coding practices at different hospitals to result in significantly different results that what they have observed.

3. I am also concerned about the lack of racial diversity in this single center study. There is extensive evidence indicating that different racial groups have significantly different rates of ACPs and DNR orders when inpatient. Specifically, patients who identify as black are far less likely to have a DNR order when hospitalized. Therefore, a population with greater racial diversity may have very different performance metrics for iDNR vs eDNR. The authors should address this issue in the discussion.

4. The authors use and presentation of the CCI is confusing. The CCI is a continuous score ranging from 0 to 37 based on a weighted assessment of each individual comorbidity. The authors indicate that they are using the CCI but then present data in categories of 0, 1, 2-3, and 4+. The language used in the results seems to indicate that these are not categories of the scores but rather number of comorbidities. As each comorbidity carries different weights, simply counting the number of comorbidities is misguided. Moreover, when considering advanced directives, certain comorbidities are far more associated with a DNR order (e.g. cancer) than others. Therefore, simply looking at the CCI to compare differences between the iDNR and eDNR group can lead to misguided conclusions. I would suggest the authors clarify how the CCI is being used in the models and would present different rates of certain key comorbidities like cancer.

5. I am additionally confused by the use of cost as a potential outcome. The fact that patients with a DNR order have lower costs is not surprising or informative. Patients with a DNR order are less likely to receive high cost invasive therapies and, given associations with mortality, have shorter length of stays. Moreover, I am also confused by the primary exposure in the cost analysis as presented in Table 4. It appears that the primary exposure is a 4 level categorical variable with levels of Died, DNR, Died & DNR, and Lived & no DNR. This variable is very confusing as there are potentially overlapping categories and potentially groups not represented. For example, if a patient died but had no DNR does that mean they are level 1 or is there overlap between level 1 and level 3. Moreover, if the groups are mutually exclusive, is level 2 really Lived and DNR? The authors need to clarify why cost is an important outcome and how they are defining the exposure. One might consider looking at LOS rather than cost given that costs can vary widely between hospitals.

6. Several studies have shown significant changes to coding patterns associated with the shift from ICD-9 to ICD-10 in October 2015.While the authors address the validity of iDNR in several subgroups, they do not report differences in iDNR validity for ICD-9 vs ICD-10 codes. I would strongly suggest the authors do further subgroup analysis focusing on ICD-9 vs ICD-10 to show that secular changes in coding pratices and the shift from ICD-9 to ICD-10 has not changed validity.

Minor Points

1. In the introduction, the authors indicate that DNR is a legal document. However, in most states it is a hospital order that only applies to that hospitalization. Medical Orders for Life Sustaining Treatments (MOLST) forms have taken over in many states as the primary method of communicating goals of care between hospitalizations. In most states, MOLST forms are legal documents. The authors should just clarify this point about DNR orders.

2. The Methods Data Source section states that ‘both authors had full access to all the data’. There are 3 authors so I would suggest modifying the text to indicate whether all others, 2/3 authors, etc had access to the data.

3. In the results discussing mortality, the section around non-DNR drivers of mortality is a bit distracting. Knowing that “other pay sources” had the highest mortality is not related to the central question of iDNR vs eDNR.

4. Figure 1 is really a Table, not a figure.

5. Previous work has shown that DNR status can be a major confounder in comparative effectiveness and observational research (Bradford et al. CCM. 2014). Being able to identify DNR status in large administrative databases would enable to researchers to adjust for this key confounder if/when necessary. The authors should explore this fact in the discussion.

6. PLOS authors have the option to publish the peer review history of their article (what does this mean?). If published, this will include your full peer review and any attached files.

Reviewer #1: No

---

## [Author Response · Author response to Decision Letter 0]

23 Nov 2022

Response to Reviewers 

Journal Requirements:

We have made the following changes – based on PLOS ONE - Formatting Sample Main Body PDF:

Changed all headings (level 1) to font size 18

Changed all subheading (level 2) to font size 16

Changed heading of “Methods” to “Materials and methods” 

Changed all subheading to be written in sentence case

Moved tables to be included directly after the paragraph in which they are first cited

Changed Table title to bold font

Changed references titles to be in sentence case

Change reference journals names to be abbreviated 

We have also made the following changes – based on PLOS ONE - Formatting Sample Title Authors Affiliations PDF:

Title changed to sentence case

Changed author names to not include education, added provided symbols as necessary

Added author contribution

Participant consent was waived by the Pennsylvania State University Institutional Review Board. This is now noted in the Methods section.

We received the data in a fully de-identified fashion; this is now mentioned in the Methods section. As mentioned above, informed consent was waived by the Pennsylvania State University Institutional Review Board. 

3. We note you have included a table to which you do not refer in the text of your manuscript. Please ensure that you refer to Table 2 in your text; if accepted, production will need this reference to link the reader to the Table.

Table 2 is now referenced in the text, and it has been moved to appear directly under the paragraph where it is referenced. 

4. Please include your tables as part of your main manuscript and remove the individual files. Please note that supplementary tables (should remain/ be uploaded) as separate "supporting information" files

All tables have been included as part of the main manuscript, directly under the paragraph where they are mentioned in the text. 

Additional Editor Comments:

Reviewer 1 has pointed out several important factors to address. Authors are urged to take a more cautious approach to interpreting findings and stating conclusions. Please address each of the reviewer's comments and suggestions. If data is available, suggestions for expanding age range and ethnic diversity, etc. appear reasonable. The treatment of continuous variables and unweighted variables needs to be addressed according to PLOS' requirements for analyses to be conducted according to high standards. This is a valuable work, and we look forward to your response.

Each of Reviewer comments has been addressed as we describe below. We appreciate the expression of the value of the work.

Reviewer Comments:

Major Points

1. My primary concern regarding the manuscript is the disconnect between the generalizability with which the authors discuss their results and the restrictive nature of their cohort. The authors use their results to discuss how discharge and claims data can be used to identify DNR orders in future studies. However, their cohort is exceedingly restrictive and may not be the most appropriate with which to identify patients admitted with heart failure. 

Language throughout the manuscript has been changed to underscore the restrictive nature of the cohort and the limited generalizability. Specifically, this has been done in the “Conclusions” section of the “Abstract”, as well as the first and third paragraph of the “Discussion”, and the “Conclusions” section. 

Their reported statistics of sensitivity, specificity, PPV, and NPV are only valid for this specific cohort and likely are not generalizable to other conditions or patient cohorts. The authors should first consider their patient selection along the below delineated lines and also discuss their results with much greater clarity about the limitations of their findings throughout the abstract and manuscript, not just in the limitations section of the discussion.

a. Age – the authors select patients who are >65 years of age. Use of DNR orders does increase with age meaning there would be greater prevalence of DNR orders in an older cohort. However, previous studies have indicated use of DNR orders begins to increase at around age 40. Use of an older cohort would result in higher sensitivity and specificity due to the higher prevalence. The authors should address this by either widening the cohort or doing a sensitivity analysis which includes patients down to age 40.

We have revised our language to be more cautious about generalizability beyond a similar patient population. We no longer have access to the EMR records of the sample of patients, therefore we would not be able to expand the cohort or do a sensitivity analysis. In the “Limitations” section of the “Discussion” we have added that in the future it would be useful to look at patients aged 40 and above.

b. Cohort definition – the authors only include patients with a principle diagnosis of HF (ICD9 and ICD10) as the ‘HF’ cohort. However, previous work (Mehta et al. Journal of Critical Care. 2015.) demonstrates that the most severely ill patients with HF (i.e. those most likely to have a DNR order) may have a principle diagnosis of acute respiratory failure and a secondary diagnosis of HF (i.e. presented with hypoxia due to pulmonary edema) or a principle diagnosis of acute MI with a secondary diagnosis of HF. These alterative coding pathways may be used to increase reimbursement but still tend to identify patients with HF. The authors could consider a sensitivity analysis with a wider cohort definition or they should mention that their definition may have selectively excluded individuals most likely to have a DNR order. 

This is an important point. We agree that our cohort identification strategy may have left out some patients with severe HF that have progressed to other conditions such as ARF and MI who might be discharged with a primary diagnosis of ARF or MI. However, attempting to include patients with a secondary diagnosis of HF and primary diagnosis of ARF or MI would also capture patients for whom HF was not necessarily severe and was a comorbid condition. It seems to us that only a prospective study could adequately distinguish between these types of patients. We have opted to continue with our cohort as defined, but have added language to the limitations section acknowledging this potential selection bias and appreciate the reviewer for pointing it out. 

c. Choice of heart failure – the authors restrict their analysis to HF only for unclear reasons. They suggest that studies that include all patients may include diagnoses for which DNR may not be relevant. However, DNR orders issued early in the course of admission tend to be more indicative of comorbid condition rather than the acute presenting illness. Moreover, other cohorts such as cancer may be even more relevant for DNR orders. The authors should provide greater justification for why HF was chosen, what biases such a choice may create, and limit their discussion about the utility of their findings to patients with HF.

We have clarified in the second paragraph of the “Introduction” section that HF was chosen because:

It can be important for patients to define their resuscitation preferences as CPR is often not beneficial for individuals at end-of-life (EOL) who are already weak and frail, such as those with severe chronic and terminal illnesses such as heart failure (HF). 

Specific statistics of HF and its trajectory make communication during this time even more important 

2. My second major concern deals with the use of a single center and conclusion that can be drawn from it. I commend the authors on their sample size and efforts but I am concerned about any conclusions drawn from a single center on generalizable billing practices that might be heavily biased. The authors correctly mention several possible reasons why DNR orders may be systematically over or under estimated in billing codes from a given center. Each health center tends to bill in a unique fashion. Biases in terms of including or not including certain codes tend to happen at the hospital level, not the patient level. Therefore, the use of a single health system does not necessarily reduce the potential of coding bias to either over or under represent DNR orders in ICD codes. The authors should be far more specific in addressing the limitations of a single center in their discussion. Specifically, they should address the potential for different coding practices at different hospitals to result in significantly different results that what they have observed. 

This has been specifically addressed in the first paragraph of “Limitations” in the “Discussion” section.

3. I am also concerned about the lack of racial diversity in this single center study. There is extensive evidence indicating that different racial groups have significantly different rates of ACPs and DNR orders when inpatient. Specifically, patients who identify as black are far less likely to have a DNR order when hospitalized. Therefore, a population with greater racial diversity may have very different performance metrics for iDNR vs eDNR. The authors should address this issue in the discussion.

We agree; the cohort is reflective of the regional population in central Pennsylvania, and is not very diverse. We have added language acknowledging this limitation in the Discussion section. 

4. The authors use and presentation of the CCI is confusing. The CCI is a continuous score ranging from 0 to 37 based on a weighted assessment of each individual comorbidity. The authors indicate that they are using the CCI but then present data in categories of 0, 1, 2-3, and 4+. The language used in the results seems to indicate that these are not categories of the scores but rather number of comorbidities. As each comorbidity carries different weights, simply counting the number of comorbidities is misguided. 

We agree that the language implied that these were numbers of comorbidities. We have changed the language in the “Results” section to be consistent with our treatment of CCI as categories of an index. 

5. Moreover, when considering advanced directives, certain comorbidities are far more associated with a DNR order (e.g. cancer) than others. Therefore, simply looking at the CCI to compare differences between the iDNR and eDNR group can lead to misguided conclusions. I would suggest the authors clarify how the CCI is being used in the models and would present different rates of certain key comorbidities like cancer. 

The CCI already assigns greater weight to comorbidities that carry a higher mortality risk. We ran some sensitivity analyses with a few of the more severe comorbidities (e.g., metastatic cancer) in the index and found that they were not more informative than the index overall. We therefore opted to include the index as a way to reflect overall comorbidity burden. This is addressed in the “Covariates” portion of the “Material and methods” section. 

6. I am additionally confused by the use of cost as a potential outcome. The fact that patients with a DNR order have lower costs is not surprising or informative. Patients with a DNR order are less likely to receive high cost invasive therapies and, given associations with mortality, have shorter length of stays. 

Moreover, I am also confused by the primary exposure in the cost analysis as presented in Table 4. It appears that the primary exposure is a 4 level categorical variable with levels of Died, DNR, Died & DNR, and Lived & no DNR. This variable is very confusing as there are potentially overlapping categories and potentially groups not represented. For example, if a patient died but had no DNR does that mean they are level 1 or is there overlap between level 1 and level 3. Moreover, if the groups are mutually exclusive, is level 2 really Lived and DNR? 

The authors need to clarify why cost is an important outcome and how they are defining the exposure. One might consider looking at LOS rather than cost given that costs can vary widely between hospitals. 

Regarding cost as a potential outcome, we disagree somewhat with the reviewer. Numerous studies have suggested that much of the resources used in the care of people in the last year of life are futile. Patients with DNR forego potentially futile care, consistent with their wishes. Using cost as an outcomes provides some quantification of those benefits. Also, a DNR is a piece of paper that expresses a patient’s wish not to receive CPR in the event of cardiac arrest. No other care should be impacted by the presence of a DNR. Patients should receive all therapies, including high cost therapies, as long as there is no cardiac event. So, the DNR is not inherently cost-saving, unless the patient has a cardiac event and the family and clinical team honor the patient’s wishes. The use of main effects plus an interaction term for DNR and Died are intended to reflect this hypothesis about cost savings and DNR. We should expect that a patient who has a DNR and dies should have cost savings since they should not receive heroic care if the death was from a cardiac event. Patients who do not have a DNR and who also die should be more costly since they should receive heroic measures. Patients who do not have a DNR and who live should have costs that are similar to patients who have a DNR and live. The combination of main effects plus interaction term captures all of these groups, with no overlap and no double counting. We agree with the reviewer, however, that our inclusion of Lived & DNR as a reference group is confusing, would imply overlap and double counting, and detracts from the main effects plus interaction approach. This reference group has been removed. Also, since this is a single center study there should be no concerns about costs varying widely between hospitals, though of course generalizability is still a concern as mentioned above.

7. Several studies have shown significant changes to coding patterns associated with the shift from ICD-9 to ICD-10 in October 2015.While the authors address the validity of iDNR in several subgroups, they do not report differences in iDNR validity for ICD-9 vs ICD-10 codes. I would strongly suggest the authors do further subgroup analysis focusing on ICD-9 vs ICD-10 to show that secular changes in coding pratices and the shift from ICD-9 to ICD-10 has not changed validity. 

Agreed, the further subgroup analysis focusing on ICD-9 vs ICD-10 is now shown in Table 3. 

Minor Points

1. In the introduction, the authors indicate that DNR is a legal document. However, in most states it is a hospital order that only applies to that hospitalization. Medical Orders for Life Sustaining Treatments (MOLST) forms have taken over in many states as the primary method of communicating goals of care between hospitalizations. In most states, MOLST forms are legal documents. The authors should just clarify this point about DNR orders. 

We agree, and have added language in the “Introduction” section to clarify this point about DNR orders and MOLST forms. 

2. The Methods Data Source section states that ‘both authors had full access to all the data’. There are 3 authors so I would suggest modifying the text to indicate whether all others, 2/3 authors, etc had access to the data. 

Agreed, this had been updated, and we have added language in the “Data Source” section to reflect that two of the authors had access to the data. 

3. In the results discussing mortality, the section around non-DNR drivers of mortality is a bit distracting. Knowing that “other pay sources” had the highest mortality is not related to the central question of iDNR vs eDNR. 

We agreed; this has been removed from the “Results” section discussing mortality. 

4. Figure 1 is really a Table, not a figure.

Agreed, Figure 1 has been changed to Table 2, and subsequent tables have been re-numbered. 

5. Previous work has shown that DNR status can be a major confounder in comparative effectiveness and observational research (Bradford et al. CCM. 2014). Being able to identify DNR status in large administrative databases would enable to researchers to adjust for this key confounder if/when necessary. The authors should explore this fact in the discussion. 

This had been added to the “Discussion” section.

---

## [Decision Letter · Decision Letter 1]

4 Jan 2023

PONE-D-22-21926R1Validity of ICD codes to identify do-not-resuscitate ordersPLOS ONE

Dear Dr. Callahan,

Thank you for submitting your manuscript to PLOS ONE. After careful consideration, we feel that it has merit but does not fully meet PLOS ONE’s publication criteria as it currently stands. Therefore, we invite you to submit a revised version of the manuscript that addresses the points raised during the review process.

We look forward to receiving your revised manuscript.

Kind regards,

Keith M. Harris, PhD

Academic Editor

PLOS ONE

Journal Requirements:

Additional Editor Comments:

Thank you for resubmitting an improved manuscript. There are just a few points to address. Please take all of Reviewer 1's points carefully and address each. None of these impact on the validity of the study, but rather the validity of the language used to describe the study. Improving the accuracy and completeness will be helpful for researchers interested in DNROs. PLoS is generous with word counts, so feel free to explain these matters.

Reviewers' comments:

Reviewer's Responses to Questions

**Comments to the Author**

1. If the authors have adequately addressed your comments raised in a previous round of review and you feel that this manuscript is now acceptable for publication, you may indicate that here to bypass the “Comments to the Author” section, enter your conflict of interest statement in the “Confidential to Editor” section, and submit your "Accept" recommendation.

Reviewer #1: (No Response)

2. Is the manuscript technically sound, and do the data support the conclusions?

Reviewer #1: No

3. Has the statistical analysis been performed appropriately and rigorously? 

Reviewer #1: Yes

4. Have the authors made all data underlying the findings in their manuscript fully available?

Reviewer #1: Yes

5. Is the manuscript presented in an intelligible fashion and written in standard English?

Reviewer #1: Yes

6. Review Comments to the Author

Reviewer #1: I commend the authors for their responses to my previously submitted review. While they have improved many aspects of the manuscript, I continue to have significant reservations about the generalizability of the findings and I do not believe the authors have narrowed their conclusions and interpretation of their data. Based on their paper, iDNR may be a valid proxy for eDNR in their specific hospital for older patients with heart failure in a predominantly white population. As coding patterns vary from hospital to hospital and from one condition to another, they cannot conclude that iDNR will perform as well in another hospital or health care system. Moreover, PPV and NPV will change based on prevalence and a hospital with more non-White patients will have a very different prevalence of eDNR. The authors acknowledge this limitation in their response but I believe that the current presentation might be misleading. I would recommend two specific changes to ensure readers have a clear picture of the scope of the study. I would suggest:

1. Title - Validity of ICD codes to identify do-not-resuscitate orders. This title very much implies a generalization of the findings that is not appropriate. I would suggest a modification to something along the lines of: "Validity of ICD-codes to identify do-not-resuscitate orders among older adults with heart failure: A Single Center Study"

2. Abstract. Conclusions. "ICD codes provide a reasonable proxy for DNR orders." I would strongly argue that the authors do not have sufficient evidence to say that iDNR is a reasonable proxy for eDNR for all patients. As many readers only read the abstract, the abstract needs to be clear about the implication. I would suggest "ICD codes provide a reasonable proxy for DNR orders among hospitalized older adults with heart failure. Further research is necessary to determine if billing codes can identify DNR orders in other populations."

Minor Point

In Table 3, to more clearly state that the CCI is a score not a count of comorbidities i would change the row title to "Charlson Comorbidity Index Score"

7. PLOS authors have the option to publish the peer review history of their article (what does this mean?). If published, this will include your full peer review and any attached files.

Reviewer #1: No

---

## [Author Response · Author response to Decision Letter 1]

31 Jan 2023

Comments to the Author

Reviewer #1: I commend the authors for their responses to my previously submitted review. While they have improved many aspects of the manuscript, I continue to have significant reservations about the generalizability of the findings and I do not believe the authors have narrowed their conclusions and interpretation of their data. 

Thank you for your time reviewing the paper and making sure the findings of our study are communicated in the most appropriate way. Below we address all of concerns your review revealed. 

Based on their paper, iDNR may be a valid proxy for eDNR in their specific hospital for older patients with heart failure in a predominantly white population. As coding patterns vary from hospital to hospital and from one condition to another, they cannot conclude that iDNR will perform as well in another hospital or health care system. Moreover, PPV and NPV will change based on prevalence and a hospital with more non-White patients will have a very different prevalence of eDNR. The authors acknowledge this limitation in their response but I believe that the current presentation might be misleading. I would recommend two specific changes to ensure readers have a clear picture of the scope of the study. I would suggest:

1. Title - Validity of ICD codes to identify do-not-resuscitate orders. This title very much implies a generalization of the findings that is not appropriate. I would suggest a modification to something along the lines of: "Validity of ICD-codes to identify do-not-resuscitate orders among older adults with heart failure: A Single Center Study"

We agree, the additions to the title will add further clarity. Language has been added to the title. 

2. Abstract. Conclusions. "ICD codes provide a reasonable proxy for DNR orders." I would strongly argue that the authors do not have sufficient evidence to say that iDNR is a reasonable proxy for eDNR for all patients. As many readers only read the abstract, the abstract needs to be clear about the implication. I would suggest "ICD codes provide a reasonable proxy for DNR orders among hospitalized older adults with heart failure. Further research is necessary to determine if billing codes can identify DNR orders in other populations."

We agree, the language presented in the conclusion of the Abstract has been changed to reflect the wording suggested. 

Minor Point

In Table 3, to more clearly state that the CCI is a score not a count of comorbidities i would change the row title to "Charlson Comorbidity Index Score"

This row title has been changed to the suggested in all tables.

---

## [Editor Report · Decision Letter 2]

1 Mar 2023

Validity of ICD codes to identify do-not-resuscitate orders among older adults with heart failure: A single center study

PONE-D-22-21926R2

Dear Dr. Callahan,

We’re pleased to inform you that your manuscript has been judged scientifically suitable for publication and will be formally accepted for publication once it meets all outstanding technical requirements.

Kind regards,

Keith M. Harris, PhD

Academic Editor

PLOS ONE
---

## [Editor Report · Acceptance letter]

3 Mar 2023

PONE-D-22-21926R2 

Validity of ICD codes to identify do-not-resuscitate orders among older adults with heart failure: A single center study 

Dear Dr. Callahan:

I'm pleased to inform you that your manuscript has been deemed suitable for publication in PLOS ONE. Congratulations! Your manuscript is now with our production department. 

Kind regards, 

on behalf of

Dr. Keith M. Harris 

Academic Editor

PLOS ONE